# Gut Microbiome of Healthy and Arthritic Dogs

**DOI:** 10.3390/vetsci7030092

**Published:** 2020-07-14

**Authors:** Michela Cintio, Elisa Scarsella, Sandy Sgorlon, Misa Sandri, Bruno Stefanon

**Affiliations:** Department of Agriculture, Food, Environmental and Animal Science, University of Udine, 33100 Udine, Italy; michela.cintio@uniud.it (M.C.); scarsella.elisa@spes.uniud.it (E.S.); sandy.sgorlon@uniud.it (S.S.); misa.sandri@uniud.it (M.S.)

**Keywords:** gut microbiome, hematological parameters, arthritis, dogs

## Abstract

Several studies have underlined the interplay among host-microbiome and pathophysiological conditions of animals. Research has also focused specifically on whether and how changes in the gut microbiome have provoked the occurrence of pathological phenomena affecting cartilage and joints in humans and in laboratory animals. Here, we tried to evaluate the relationship between the gut microbiome and the hip and elbow arthritis in owned dogs. The study included 14 dogs suffering from chronic arthritis (AD) and 13 healthy dogs (HD). After the first visit and during the period of the study, the dogs, under the supervision of the owner, were fed a semi-moist complete diet supplemented with omega 3 fatty acids. Feces and blood samples were collected in the clinic at the first visit (T0) and after days (T45). The plasma C-reactive protein (CRP) was higher, and the serum vitamin B12 and folate concentrations were lower (*p <* 0.05) in the AD group in comparison to the HD group. Data of the fecal microbiome showed that the relative abundances of the genus *Megamonas* were higher in AD (*p <* 0.001), while the relative abundance of the families Paraprevotellaceae, Porphyromonadaceae, and Mogibacteriaceae was significantly lower in comparison to HD. The results of the study identified several bacterial groups that differed significantly in the fecal microbiome between healthy and diseased dogs. If the observed differences in fecal bacterial composition predispose dogs to hip and elbow arthritis or if these differences reflect a correlation with these conditions deserves further investigation.

## 1. Introduction

In the last decade, several studies have focused on the gut microbiome in dogs and on the relationship that inflammatory bowel diseases and intestinal dysbiosis have with pathological conditions of the organism [1,2]. It has been also demonstrated, both in human and animal models, that the microorganism inhabitants of the gut and the compounds they produce, such as short-chain fatty acids, lactic acid, and metabolites of amino acids, affect host physiology [3]. Among the others, the microbial metabolism of tryptophan and threonine and the related interaction with the availability of neurotransmitters for the host has led to the development of the brain-gut-microbiome axis concept [4,5]. It has also been reported that the microbiome can influence distant organs by three principal effects, regulating the nutritional absorption and production of vitamins, regulating the immune system, and translocating bacteria through the endothelial barrier and into the bloodstream [6]. This has been investigated in human and laboratory animals [7,8,9,10], but scientific pieces of evidence are still lacking for dogs.

Recent studies have brought to light a possible new interplay of the gut microbiome with pathological conditions other than inflammatory bowel diseases, such as metabolic syndrome and joint damage, the latter caused by chronic low-grade systemic inflammation [3]. It is already accepted that when intestinal dysbiosis occurs, the whole organism suffers from several metabolic and inflammatory diseases, and in this context, it is conceivable to assume that also the pathogenesis of the joint disease could be related to it.

According to Tomasello et al. (2014) [11], in humans affected by irritable bowel syndrome, it was observed that during microbial dysbiosis, an immune system dysregulation could occur, which consequently brought to a local inflammatory state. Hernandez et al. (2018) [6] reported that during dysbiosis, a leaky gut could happen, and this allowed a transfer of gut microbes from the intestinal lumen to other sites of the body (e.g., mammary gland; [12]) through dendritic cells (DCs) and macrophages. If this crossing is implicated in the onset of joint disease is a hypothesis that deserves further investigation.

Osteoarthritis is a chronic disease that in the long run leads to a degradation of cartilage and bone, abnormal growth of bone tissue (osteophytes), and inflammation of the synovial membrane (synovitis), causing arthrosis and pain, stiffness, and loss of functionality of the joints [13,14,15]. This pathology is multifactorial and is mainly related to genetic predisposition, mechanical events, age, sex, lifestyle, poor body condition, and excess body weight [3].

A common feature of arthrosis is local and sometimes systemic inflammation, but, until now, the gut-joint axis has only been hypothesized in human and laboratory animals [9,10,11,12,13,14,15,16], considering that clinical evidence linking gut microbiome with arthritis is still lacking. In this study, we evaluated hematological parameters, fecal microbiome composition, and end products of fermentation in healthy and arthritic dogs with the aim to evaluate a relationship, if any, supporting the microbiome-joint axis in dogs.

## 2. Materials and Methods

### 2.1. Animals and Housing

For the purpose of the study, 13 healthy dogs (HD) and 14 dogs with hip or elbow arthritis (AD) were enrolled from a veterinary local clinic. AD dogs had a history of chronic arthritis and were not under NSAIDs (non-steroidal anti-inflammatory drugs) or steroidal treatments since the last month. Arthritis was diagnosed by the veterinarians on the basis of clinical anamnesis for limping forelimbs and/or associated with posterior limping; rapid fatigue after effort; decrease in motor activity; radiographic evidence of osteoarthritis (AD) in one or two joints (elbow, hips). Dogs were of both sex and different breeds, including crossbreeds, were older than 18 months, and weighed more than 10 kg and less than 40 kg. Details of the dogs are reported in Appendix A. After the visit, HD and AD dogs were fed a semi-moist complete diet (N.P. Industries, Udine, Italy). The diet consisted of the diced potato (4.0%), potato starch (4.0%), chicken (17.8%), swine heart (17.0%), swine lung (10.0%), swine liver (7.4%), vitamins and minerals (1.25%), additives (thickeners, gelling agents; 1.8%), vegetable oil (1.25%), prebiotic (xylooligosaccharides, 0.01%), and oregano essential oil (0.0006%). Diet contained 80.0% of water, 7.2% of crude protein, 5.4% of lipids, 0.45% of crude fiber, 2.1% of ash, and 3.3% of starch. Lipids contained 5.3% of omega 3 (alpha-linolenic 3.7%, eicosapentaenoic 0.5%, and docosahexaenoic 0.3%). The diet was formulated to provide omega 3 fatty acids to support the metabolism of joints in the case of osteoarthritis. Throughout the period of study, dogs were housed in the usual home and condition and were followed by the owners, and informed consent was obtained from the owners of the dogs recruited for the study. All protocols, procedures, and the care of the animals complied with the Italian legislation on animal care and were approved by the ethical committee of the University of Udine (26/08/2019; protocol n. 7/2019).

### 2.2. Collection of Samples

The samples from HD were collected according to a scheduled check agreed with the owners, and for the AD, the samples were collected according to the veterinary opinion to monitor their clinical conditions. Blood samples were withdrawn at the first clinical examination (T0) and after 45 days (T45) from the beginning of the trial in the veterinary clinic. Fecal samples were also collected at T0 and T45 from the first evacuation of the morning by the owners, and the specimens were immediately frozen at −20 °C. Samples were delivered to the clinic and transported to the laboratory within 1 week and stored at −80 °C. For the analyses, the inner part of the stools was sampled for analysis.

### 2.3. Hematological Analysis

Blood samples were collected at the veterinary clinic in Li-heparin tubes (Terumo Europe N.V., Leuven, Belgium) and without anticoagulant tubes and then centrifuged at 3000 RPM for 20 min. Plasma and serum were transferred in 5 mL vials and immediately frozen at −20 °C. The samples were sent within 3 days to an external certified veterinary laboratory for a complete biochemical analysis of plasma, folate, vitamin B12, and trypsin-like immunoreactivity (TLI) in serum (Vetlab, Padova, Italy).

### 2.4. Fecal Short-Chain Fatty Acid and Lactic Acid Analysis

Lactic acid and short-chain fatty acids (SCFAs) (acetic, propionic, butyric, isobutyric, and isovaleric) were analyzed by HPLC [17], starting from 1 g of feces, which was diluted with 50 mL of 0.1 N H_2_SO_4_ aqueous solution and homogenized for 15 min by a mechanical stirrer (Instruments Srl, Milano, Italy). After centrifugation at 20,000× *g* for 20 min at 4 °C to separate the liquid phase from the solid residuals, the supernatant was filtered with a 0.45 µm syringe filter of polypore (Alltech, Casalecchio di Reno, BO, Italy). An aliquot of 20 µL of the resulting sample was injected in the HPLC equipped with an Aminex HPX-87H (Bio-Rad, Hercules, CA, USA) ion exclusion column (300 mm × 7.8 mm, 9 µm) and a pre-column (Bio-Rad, Hercules, CA, USA) kept at 40 °C. The isocratic elution flux was 0.6 mL/min, using 0.008 N H_2_SO_4_ solution as a mobile phase, and the detection length was 220 nm. SCFAs and lactic acid concentrations were calculated with reference to a standard solution of 4.50 mg/mL of lactic acid, 5.40 mg/mL of acetic acid, 5.76 mg/mL of propionic acid, 7.02 mg/mL of butyric acid and isobutyric acid and isovaleric acid in 0.1N H_2_SO_4_ (Sigma–Aldrich^®^ Co., Milano, Italy). Quantifications were calculated using an external calibration curve based on these standards. Each acid was expressed as a molar percentage of the sum of SCFAs and lactic acid (TA).

### 2.5. Fecal DNA Extraction, Sequencing, and Taxonomic Annotation

Microbial DNA from the inner part of the stool was extracted from 150 mg of starting material using a Fecal DNA MiniPrep kit (Zymo Research; Irvine, CA, USA), following the manufacturer’s instruction, including a bead-beating step. DNA concentration was measured with a QubitTM 3 Fluorometer (Thermo Scientific; Waltham, MA, USA). The DNA was fragmented, and the 16S rRNA of V3 and V4 regions amplified for library preparation, adding also the indexes for sequencing, using a Nextera DNA Library Prep kit (Illumina; San Diego, CA, USA), with the primers suggested by Klindworth et al. (2016) [18]. The amplicons were sequenced with a MiSeq (Illumina; San Diego, CA, USA) in 2 × 300 paired-end mode.

The Quantitative Insights Into Microbial Ecology (QIIME 2) [19] was used to process the raw sequences, which were uploaded to the NCBI Sequence Read Archive (BioProject ID PRJNA611632). After demultiplexing, reads passing with Phred score ≥30 were annotated for 16S rRNA against the Greengenes database (version gg.13_8.otus.tar.gz), with 99% identify with reference sequences. Chimeras were also detected and then filtered from the reads, and the remaining sequences were clustered into operational taxonomic units (OTUs) by using an open reference approach in QIIME 2.

### 2.6. Statistical Analysis

Between samples, the minimum number of reads count was 10,578, and average reads were 36,395 ± 11,284 and 31,287 ± 16,035 for AD and HD groups, respectively (*p >* 0.05). The 16S rRNA annotated sequences were then normalized to ‰ abundance profiles for each sample and each taxonomic level, already known as relative abundance (RA). Taxa with RA lower than 10‰ were excluded from the statistical analysis [5,6,7,8,9,10,11,12,13,14,15,16,17,18,19,20].

Before statistical analysis, normality of distribution of the independent variables (SCFAs, lactic acid, and hematological parameters) was checked with the non-parametric Kolmogorov–Smirnoff test. When data were not normally distributed, a natural logarithmic transformation was used. Data were then rechecked and resulted normally distributed. The linear mixed model was used to analyze all these variables. The model included the fixed effect of time of sampling (2 levels, T0 and T45), status (2 levels AD, HD), and the interaction of time of sampling with status, with the subject (dog) as random factor repeated over the time of sampling. Bonferroni multiple testing correction was used as a significance test. For the microbiome, Shannon and Evenness diversity indices were calculated at the family and genus levels [17]. Beta diversity was assessed with the Brian Curtis dissimilarity matrix and visualized using principal coordinate analysis (PCoA) plot. Analysis of similarity (ANOSIM) was performed with the ‘Vegan’ package in R (Version 3.2.1) to test whether the microbiome significantly differed between AD and HD at T0 and T45. All these analyses were performed with XLSTAT [21]. A linear discriminant analysis (LDA) effect size (LEfSe) was applied to detect taxa that differed between diseased and healthy groups at T0 and T45 [22].

## 3. Results

Statistical analyses of plasma biochemistry results are depicted in Table 1. C-reactive protein (CRP), urea, albumin, and lipase were significantly higher in AD (*p <* 0.05) in comparison to HD. Creatine showed an increase with time of sampling (*p <* 0.05) in both dog groups, while for total bilirubin, a negative variation at T45 compared to T0 (*p <* 0.05) was registered. Besides, this latter parameter was higher in HD than AD (0.19 and 0.16, respectively; *p <* 0.05). Cholesterol underwent positive variation for the effect of time and was higher in AD than in HD (*p <* 0.05). There was a significant increase in alanine transaminase (ALT/GTP) in AD (*p <* 0.05), but it decreased with time (*p <* 0.001), while for aspartate transaminase (AST/GOT), a decrease from T0 to T45 was reported (*p <* 0.001). Alkaline phosphatase (ALP) significantly decreased from T0 to T45 (*p <* 0.05), being lower in HD, similar to the creatine kinase (CK) (*p <* 0.05). Gamma-glutamyl transferase (GGT) was higher at T45 (*p <* 0.05) compared to T0. Regarding the mineral concentrations in plasma, the analysis showed a slight increment of Ca and Cl for the effect of time (*p <* 0.001), while K and Mg were statistically higher in HD than AD (*p <* 0.05). Inorganic P registered a variation for the interaction time X status (*p <* 0.05). Na increased from T0 to T45 (*p <* 0.001), being significantly higher in HD (*p <* 0.05). Finally, the plasma concentrations of vitamin B12 and folates were significantly higher in AD (*p <* 0.05) in comparison to HD.

The concentrations and the molar proportion (%) of the SCFAs and lactic acid are reported in Appendix A. The molar concentrations of SCFAs and TA were not significant for any effect. Lactate and isovalerate showed a numerical increase in diseased subjects. The molar proportion of acetate showed a decrease (*p <* 0.05) at T45 compared to T0 in all the groups, while the molar proportion of butyrate was higher in AD in comparison to HD (7.7% vs. 7.0%, *p <* 0.05). No significant variation was registered for the other SCFAs.

Shannon index of diversity (H’) and Evenness index (J’) of the fecal microbiome significantly differed between AD and HD at the family level but not at the genus level (Figure 1).

The principal coordinate analyses (PCoA) was used to visualize beta diversity (Figure 2) between groups. The ANOSIM computed for the two times of sampling was significantly different at the family level (*p <* 0.01) between the two groups of dogs.

The differences in taxonomical terms between the two groups are depicted in Figure 3. The bacterial taxa that were significantly increased in HD dogs were the families Paraprevotellaceae, Porphyromonadaceae, and Mogibacteriaceae and the genera *Parabacteroides*, *Phascolarctobacterium,* and *p_75_a5*. Even the family Peptococcaceae and its genus *Peptococcus* together with the family Succinivibrionaceae and its genus *Anaerobiospirillum* were significantly higher in HD. Indeed, the genus *Megamonas* showed a significantly higher relative abundance in AD.

## 4. Discussion

Osteoarthritis is a multifactorial and slowly progressive disease, resulting in a difficult early diagnosis through radiography or plasma biochemical analysis [23]. Different studies have revealed the relationship between cardiovascular disease, obesity, type 2 diabetes, and osteoarthritis, focusing on the role of the metabolic syndrome in provoking joint and bone damage [24,25].

Biochemical analyses of plasma are a suitable marker of inflammation (Table 1). CRP is a parameter of the acute phase of inflammation, and its detection in plasma could be useful in carrying information for the discrimination between suppurative arthritis and osteoarthritis in dogs, as demonstrated by Hillström et al. (2016) [26]. Rafiqul et al. (2006) [23] studied CRP concentration in dogs affected by stifle osteoarthritis and observed an increased concentration of CRP after 3, 6, and 9 months of experimental study. More recently, even Hindenberg et al. (2020) [27] focused on the significance of the concentration of high CRP (>1 mg/dL) in dogs. Classifying dogs in categories based on etiology and organ system affection, they found that the prevalent etiology was the inflammation (59%), in which 51% was infectious, and interestingly, of this 51%, the 38% had a bacterial origin. Instead, relying on the affected organs, aside from multiple organs and trauma (39 and 20%, respectively), they detected an incidence of 14% for the gastrointestinal tract and 7% of the musculoskeletal system. Here, the AD group presented significantly higher values of CRP with respect to HD, and this could support that this parameter can be a marker of the arthritic process, including that of bacterial origin.

In this study, cholesterol significantly increased in AD at T45 when compared to HD, suggesting that an inflammation process was in the act. Rafiqul et al. (2006) [23] registered an increase in plasma cholesterol in dogs affected by stifle osteoarthritis. Cholesterol concentration has been reported to be an important biomarker of inflammation in obese dogs [28], mainly due to excess in the production of adipokines, which affects insulin resistance, and to the adipocyte secretions of inflammatory mediators, such as tumor necrosis factor (TNF), interleukin-6 (IL-6), and others, which contribute to systemic inflammation. Chronic low-grade systemic inflammation plays a fundamental role in the onset of several diseases in humans, such as coronary artery disease, insulin resistance, and metabolic syndrome [29,30,31]. A study on humans conducted by Farnaghi et al. (2017) [32] discovered that synovial fluid contains a low concentration of cholesterol compared to plasma levels, while the synovial fluid of osteoarthritic subjects has a higher amount of cholesterol and cholesterol crystals than healthy individuals. Furthermore, high plasma cholesterol can be found in overweight and obese dogs [33]. Since a correlation between gut dysbiosis and obesity/overweight has been reported [33,34], it is acceptable to assume that obesity conditions may be also related to other conditions of systemic inflammation, as arthritis.

ALP is a nonspecific phosphomonoesterase that hydrolyzes phosphate monoesters, and it is found attached to the plasma membrane, where extensive transport takes place [35]. ALP localized in the plasma membrane of the osteoblastic cells seems to have a specific role in bone mineralization [36], even if more pieces of evidence are needed to be found. ALP is found in many tissues, but especially in bones, liver, bile ducts, and in the gut epithelium, and an increase of this enzyme in plasma constitutes a biochemical marker of disease of these tissues; ALP is also used, together with epidemiologic, clinical, laboratory, and radiographic findings, to characterize polyarthritis in dogs [37]. In this study, we detected a significant decrease in ALP at T45, both in AD and more evidently in HD groups. Musco et al. (2019) [38], in a dietary intervention study on dogs affected by osteoarthritis, found lower ALP activity in the group receiving the prebiotic supplement than in sick dogs. In addition, mucosal expression, intestinal activity, and fecal concentration of ALP have been discovered to be lower in dogs with chronic inflammatory enteropathies [39], advising the use of this enzyme as a biomarker, also for enteropathies in dogs.

Cobalamin (vitamin B12) and folates (vitamin B9) detected in serum represent reliable functional biomarkers of intestinal permeability and malabsorption in dogs [40,41]. Cobalamin is absorbed only in the ileum, and its lower concentration is often associated with chronic intestinal pathologies in dogs [42,43,44], presumably manifested as small intestine malabsorption, secondary small intestine dysbiosis (extensive utilization of vitamin B12 by commensal bacteria), or both. Hypocobalaminemia is associated with lower serum albumin concentration [42] in the case of a gastrointestinal protein loss due to chronic intestinal inflammation [45,46]. However, in the present study, albumin concentration was higher in the AD group, although vitamin B12 was decreased; hence, it is not possible to attribute these changes to protein-losing enteropathy.

Folates are firstly absorbed in the duodenum and proximal jejunum as folate glutamate through B9 carriers, and they are principally produced by *Bifidobacteria* and *Lactobacillus* strains [47]. According to Heilmann et al. (2018) [44], hypofolatemia can derive from chronic malabsorption in the proximal small intestine of dogs, but at the same time, an increase or a normal concentration of vitamin B9 can be a symptom of a secondary small intestine dysbiosis, in which a high production of folates is made by resident microbiota or hypocobalaminemia [48]. Even Xu et al. (2016) [49] registered a decrease in serum folate in dogs affected by inflammatory bowel disease (IBD), together with lower cobalamin concentration. In this study, these parameters were sharply higher in HD group than in the AD group, both at T0 and T45, suggesting a healthier and more functional gut microbiome in the former dogs, able to supply the required amount of vitamins the organism needs. It is already evident the capacity of the gut population to synthesize the necessary elements to sustain the organism’s homeostasis through the production of vitamins of the B group and secondary metabolites, such as SCFAs and lactic acid [50]. Dysregulation of this mechanism could compromise the mutualistic relationship among gut microbiome and host.

In regard to SCFAs, no particular variations were detected between the two groups, except for the molar proportion of butyrate that showed an increase in the AD group (Appendix A). The lack of substantial differences between the two groups in terms of SCFAs could be consequent to the rapid absorption of these metabolites, leaving a small but detectable concentration in feces. As already known, dogs do not utilize SCFAs as the principal source of energy, as ruminants do, but the dissimilarities, at least, indicate a variation in nutrient metabolism associated with a modification of gut microbiome.

Acetate, propionate, and butyrate exert beneficial effects on the homeostasis of metabolic functions since they have anti-inflammatory properties by controlling the development and modulating the immune system [51]. Besides the fact that butyrate is apported with the diet, this acid is mainly produced by the fermentation of the dietary fiber by *Clostridial* cluster IV and XIVa [52] in the gut lumen. Moreover, it acts through G protein-coupled receptor (GPCRs) pathway or by reducing histone deacetylases (HDACs) [53], involving the recruitment of macrophages and DCs to facilitate differentiation of T reg cells and immune-regulatory IL-10 [54]. Butyrate also supplies energy to the colonocytes, which means that a significant proportion of microbial-released butyrate is rapidly taken up and consumed locally [55], regulate host cell responses, and, for this reason, is considered to exert health-promoting effects on the colon [56]. Unexpectedly, our data indicated that the molar proportion of butyrate was higher in AD, even though the mean values of butyrate proportion were numerically close between the two groups. This could be explained by the fact that arthritic dogs possess many butyrate producers, such as *Megamonas* genus, as registered in the study by Sandri et al. (2017) [17], and/or a relatively few bacterial butyrate-consumers. Moreover, since butyrate can even be synthesized from acetate and lactate from the interactions with the microbial ecosystem [57,58], the increased butyrate in AD is likely due to this over synthesizing activity.

The reduction of alpha diversity indices (Figure 1), at least at the family taxonomic level, indicated that a dysbiosis probably occurred in the AD group, as supported by Minamoto et al. (2015) [59]. Furthermore, we also found differences in the relative abundances of bacterial taxa between AD and HD (Appendix A) and beta diversity at the family level (Figure 2), suggesting a link between the microbiome and joint inflammation, even though no variation was observed at the genus level.

Although the markers of the leaky gut were not analyzed, the variation of the gut microbial community and the decrease of serum cobalamin and folates supported the hypothesis that a dysbiosis occurred. Several studies have investigated the modulation in gastrointestinal bacterial patterns and gut functional status in patients with arthritis. Ad exemplum, the study conducted by Muniz Pedrogo et al. (2019) [60] on humans reported an increased abundance of Clostridiaceae in both rheumatoid arthritis and IBD-associated arthropathy. Instead, Coulson et al. (2013) [61] found a high number of Bacteroides, *Eubacterium* (*Collinsella aerofaciens*), *Lactobacillus*, *Bifidobacterium*, *Clostridium*, Coliforms (*Escherichia coli*, *Klebsiella pneumoniae*), *Enterococcus*, *Streptococcus*, *Staphylococcus,* and *Prevotella* in patients with knee osteoarthritis. Wolfe et al. (2000) [62] reported that patients diagnosed with musculoskeletal diseases registered also common predispositions to dyspepsia, nausea, abdominal bloating, and irregular bowel habits. The interference of gut microbiota with bone metabolism has been postulated also in mice, in which a reduction of bone loss associated with ovariectomy has been reported after probiotic administration [63,64]. Even Mc Cabe et al. (2013) [65] found that the use of probiotic on male mice decreased intestinal inflammation and increased bone density. Another study on mice, supporting the relation among microbiome and arthritis, found that members of the genus *Prevotella* might influence bone loss by regulating the levels of SCFAs that mediate osteoclastogenesis in the host [66]. The differences of relative abundance of taxa in AD and HD groups found in tour study do not overlap with the previous researches, but it is legitimate to consider that a strict comparison of dogs’ microbiome with humans and mice microbiome is not feasible due to the peculiar anatomy and physiology of these species, and then the observed differences were expected.

LEfSe analysis (Figure 3) revealed deep differences between the relative abundances of some taxa. Interestingly, we found a greater abundance of the genus *Megamonas* in AD, while the other genera were higher in HD. Other than butyrate, *Megamonas* is known to produce propionate, which has been shown to possess anti-inflammatory properties, and, additionally, this genus affects the metabolic rate of the host organism [67]. Here, the relative abundance of *Megamonas* in AD was 2-fold higher than that of the HD group and vice versa, while *Phascolarctobacterium* was 2-fold lower. It seemed that a shift from *Megamonas* with the *Phascolarctobacterium* occurred in AD and HD microbiome since these taxa belong to the same family of Veilloneaceae. Similar results have also been observed in human studies [68,69], suggesting that these two genera occupy the same ecological niche and compete for the same substrates. On the other hand, it should be considered that data refer to relative abundances, meaning that the increase of *Megamonas* might be due to a decrease of other genera and so do not necessarily reflect a higher amount of this genus in the microbial population of arthritic dogs.

We also detected, in HD group, a higher abundance of bacterial members of the Clostridiales order (Mogibacteriaceae, Peptococcaceae along with *Peptococcus* genus and *Phascolarctobacterium* genus) and in Erysipelotricaceae (*p_75_a5* genus), sharing the same Firmicutes phylum. Clostridiales plays a crucial role in preventing leaky gut syndrome, which occurs when gut barrier permeability is altered, prompting excessive inflammation [70]. Similar results were found in a study on dogs conducted by Minamoto et al. (2015) [59], in which Erysipelotricaceae, Clostridiales, and Bacteroidetes were underrepresented in dogs with IBD. Even members of the Bacteroidales order (Paraprevotellaceae, Porphyromonadaceae, and genus *Parabacteroides*) showed a higher abundance in HD. On the contrary, Omori et al. (2017) [71] found a great abundance of Paraprevotellaceae and *Porphyromonas* genus in dogs affected by IBD. The presence of the *Parabacteroides* genus in humans is thought to help to prevent the invasion and colonization of pathogens by secreting bacteriocins that are toxic to other strains of bacteria [72]. Individuals with IBD often lack a population of *Parabacteroides* in their gut, suggesting that this genus also helps protect against excessive inflammation [72]. Unexpectedly, Aeromonadales—the relative belonging family Succinivibionaceae and the relative genus *Anaerobiumspirillum*—were higher in healthy dogs. These bacteria belong to the Proteobacteria phylum, which has been associated with IBD [59,60,61,62,63,64,65,66,67,68,69,70,71,72,73], even if *Anaerobiospirillum* is commonly found in feces of healthy dogs [74].

Definitively, the lesson to get from these results is that gut microbiome is a complex ecosystem, and thus the variation in the composition of the microbiome as a whole, rather than to the variation of the single taxon, can better depict the effect of an inferred factor in gut-host microbiota interaction.

The modulation of the gut microbiome by arthritis risk factors (e.g., diet, obesity, age, physical exercise, sex, genetic factors, and immunity activation) could alter the intestinal thigh junctions, allowing the crossing of the local bacteria by the DCs and macrophages, which eventually can reach tissue or organs through the bloodstream, causing an inflammatory response in the host [6]. It can be argued that the gut is a source of these microorganisms because altered gut permeability has been observed in individuals with arthritis [75,76]; therefore, some living bacteria could be transferred from the gut to the joints through the circulatory or lymphatic system [77]. The existence of this axis in dogs deserves further investigations to be better understood.

## 5. Conclusions

The gut microbiome is strictly related to several forms of pathologies. In the present study, we analyzed the variation of the gut microbial community in arthritic dogs, and the results obtained suggested that it might influence the degeneration of joints and bones through the propagation of systemic inflammation. Despite our findings on bacterial taxa of dog microbiome were not similar to those detected in other studies, we hypothesized that a large inter-individual variability among dogs exists and that different bacterial strains could contribute in a unique way to modify the inflammatory status of the subjects. Hence, it is not possible to attribute a beneficial or harmful role to a single taxon in arthritic disease but is likely that a variation of the total genera composing microbiome could reshape the entire physiology of the host.

## Figures and Tables

**Figure 1 vetsci-07-00092-f001:**
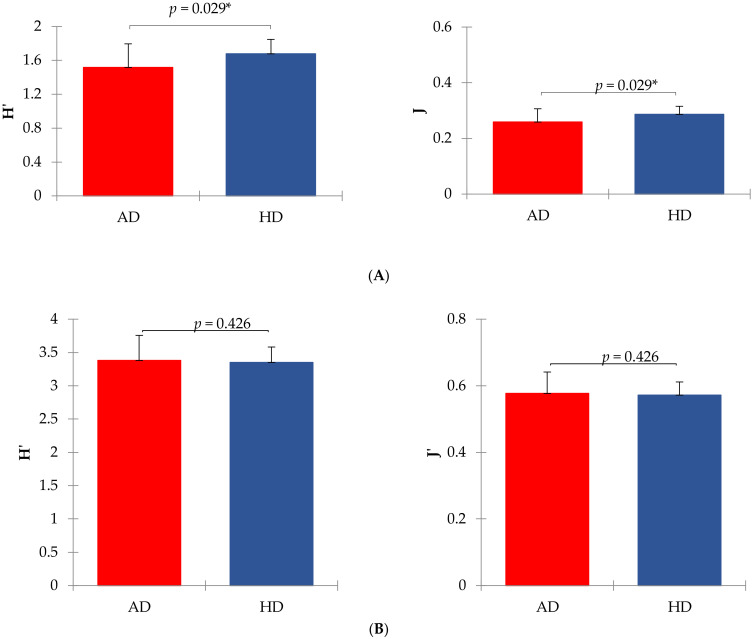
Shannon (H’) and Evenness (J’) indexes of diversity in arthritic (AD) and healthy (HD) dogs. Panel (**A**) shows mean values at the family level (means were significantly different for *p <* 0.05 between AD and HD groups). Panel (**B**) shows mean values at the genus level (means were not significantly different between AD and HD groups). Shannon index was calculated according to the equation H’ = −sum (Pi × ln Pi), where Pi = frequency of every taxon within the sample. The evenness index was calculated as J’ = H’/ln S, where S = total number of taxa within each sample.

**Figure 2 vetsci-07-00092-f002:**
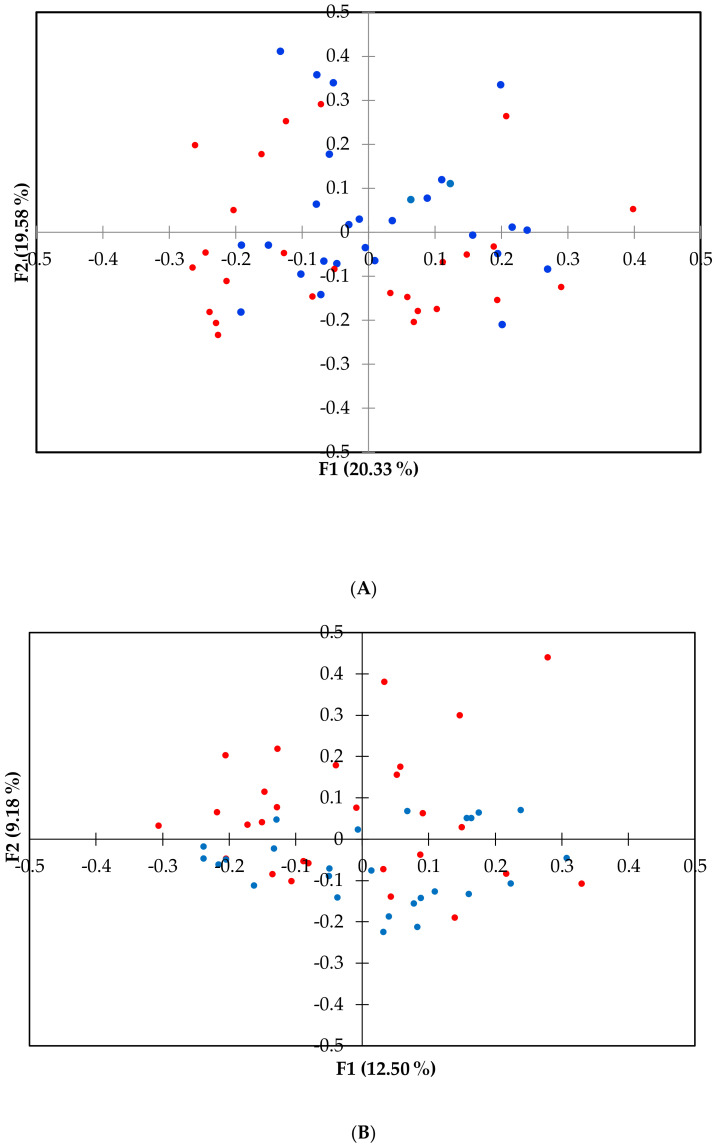
Principal coordinate (PCoA) plot representing the beta diversity of the microbial community among dogs’ health status at family (**A**) and genus (**B**) level. PCoA was calculated using the Brain Curtis dissimilarity matrix. Dots in red represent arthritic dogs (AD), and dots in blue represent the healthy dogs (HD). The analysis of similarity (ANOSIM) was significant at the family level for *p <* 0.01.

**Figure 3 vetsci-07-00092-f003:**
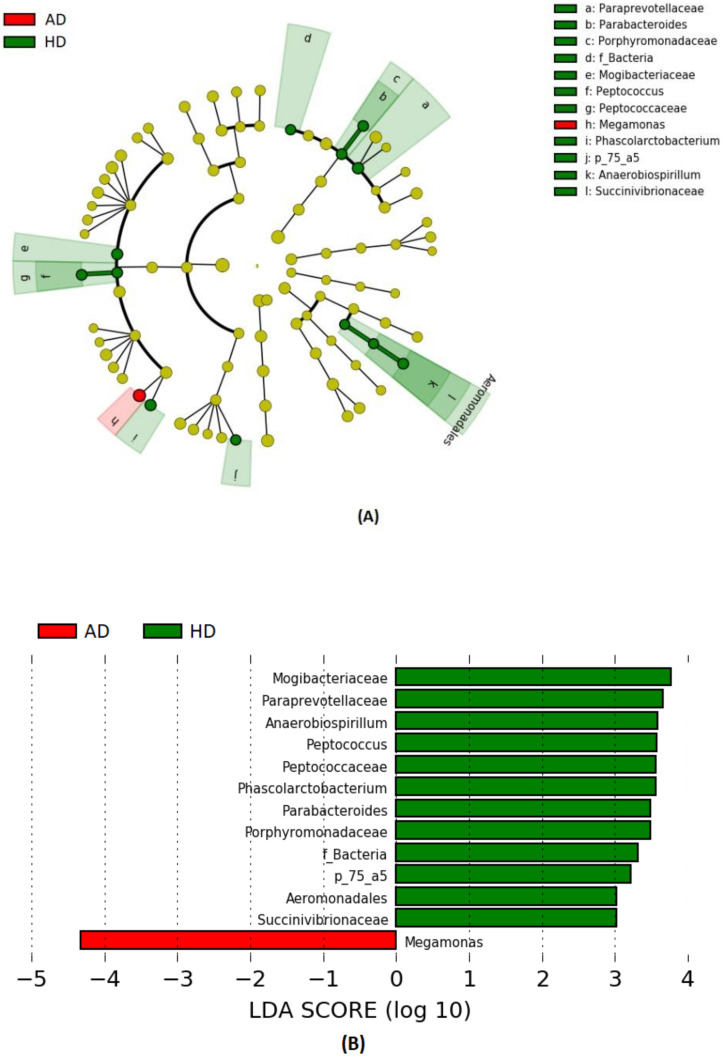
Bacterial taxa were significantly more abundant in the feces of healthy (HD) and arthritic dogs (AD). The cladogram (**A**) highlights impactful communities in individuals from each status, and (**B**) shows the score of the linear discriminant analysis (LDA, significant threshold >2).

**Table 1 vetsci-07-00092-t001:** Mean concentrations of the parameters determined by blood biochemistry in healthy (HD) and diseased (AD) dogs at the beginning of the study (T0) and after 45 days (T45).

Item	Unit	T0		T45		SEM		Effects	
		AD	HD	AD	HD		Time	Status	T x S
Glucose	mg/dL	93.14	94.23	92.86	95.77	1.20	NS	NS	NS
Fructosamine	umol/L	204.93	191.23	215.21	219.77	5.14	NS	NS	NS
Urea	mg/dL	35.50	36.28	36.71	35.08	1.38	NS	*	NS
Creatine	mg/dL	1.28	1.13	1.42	1.36	0.04	*	NS	NS
Total bilirubin	mg/dL	0.20	0.26	0.16	0.19	0.01	*	*	NS
Total proteins	g/dL	6.84	6.85	6.91	6.92	0.05	NS	NS	NS
Albumin	g/dL	3.04	2.98	3.01	2.91	0.04	NS	*	NS
Globulin	g/dL	3.81	3.87	3.90	4.02	0.06	NS	NS	NS
Albumin/Globulin	g/dL	0.81	0.78	0.77	0.74	0.02	NS	NS	NS
Cholesterol	mg/dL	238.29	228.31	266.57	242.15	7.07	NS	NS	*
Triglycerides	mg/dL	63.29	68.23	53.14	58.46	3.28	NS	NS	NS
Lipase	u/L	118.43	86.92	88.57	64.23	8.39	NS	*	NS
C-Reactive Protein	mg/dL	0.41	0.15	0.33	0.12	0.03	NS	*	NS
α-Amylase	u/L	819.71	771.77	727.07	767.92	34.51	NS	NS	NS
AST (GOT)	u/L	30.79	32.23	21.64	25.08	0.99	**	NS	NS
ALT (GPT)	u/L	56.29	29.31	31.86	22.46	2.69	**	*	*
ALP	u/L	59.57	69.62	52.14	46.69	3.15	*	NS	NS
GGT	u/L	6.93	6.46	7.07	7.46	0.20	*	NS	*
CK	u/L	102.43	158.92	90.29	97.69	8.82	NS	NS	NS
LDH	u/L	42.36	67.37	52.50	59.46	6.28	NS	NS	NS
Cholinesterase	mg/dL	6000.50	6232.00	6301.64	6472.95	231.02	NS	NS	NS
Ca	mg/dL	10.14	10.02	10.91	10.44	0.10	**	NS	NS
P-in	mg/dL	3.49	5.03	3.89	3.98	0.16	NS	NS	*
Mg	mg/dL	2.25	2.32	2.18	2.39	0.03	NS	*	NS
Fe	µg/dL	154.79	175.69	148.07	188.54	7.70	NS	NS	NS
Cl	mEq/L	109.64	109.11	111.07	110.53	0.52	*	NS	NS
K	mEq/L	4.45	4.76	4.53	4.91	0.06	NS	*	NS
Na	mEq/L	146.86	149.20	148.86	149.92	0.41	**	*	NS
TLI	ng/mL	37.74	26.61	48.62	38.25	5.37	NS	NS	NS
Folates	ug/L	10.74	11.63	11.52	12.17	0.64	NS	*	NS
Vitamin B12	pg/mL	326.64	500.69	360.79	451.46	29.07	NS	*	NS

SEM: standard error of the means; AST (GOT): aspartate transaminase; ALT (GPT): alanine transaminase; ALP: alkaline phosphatase; GGT: gamma-glutamyl transferase; CK: creatinine kinase; LDH: lactate dehydrogenase; P-in: inorganic P; TLI: trypsin-like immunoreactivity. * *p* < 0.05; ** *p* < 0.001; NS: not significant.

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
