# Peer review of "Gut Microbiome of Healthy and Arthritic Dogs"

_vetsci, 2020, doi:10.3390/vetsci7030092_

Round 1

Reviewer 1 Report

In this study on the microbiome of dogs with and without osteoarthritis, the authors test whether differences were observed in a range of blood biochemistry results and the composition of gut (fecal) bacteria. This is a relevant topic in veterinary medicine and the authors have some interesting preliminary findings with their study of a relatively small number of dogs (n=13 or 14 per group).

The main thing that should be addressed in a revision is that the authors hypothesize that a leaky gut barrier may play a role in the development of osteoarthritis. One issue with their hypothesis as stated is that the authors are not effectively describing how the loss of barrier integrity or increased epithelial permeability (“leaky gut”) is allowing increased interaction of MAMPs with the innate immune system. Their use of  “microorganisms” - which a reader would assume means viable whole microbes and therefore systemic dissemination is confusing. Most of the current literature (predominantly human) link aberrant immune reactions (Th17) leading to immune mediated osteoarthritis, rather than infectious arthritis. 

If they intend to propose and test a potential mechanism for the development of osteoarthritis that involves bacteria crossing the barrier and translocating to the joints, they need to test for microbes/microbial molecules in the affected joint tissue. Unless they document microbial translocation to the joints or can otherwise support their claims, they need to revise their hypotheses. For example - if the arthritic dogs have serum IgG specific to the implicated fecal bacteria.

I have some questions about the statistical analyses. In addition, there are some minor spelling/grammatical corrections that should be made to the manuscript. Some are listed in the suggestions below.

In the abstract, I recommend modifying this sentence:

From: The results of the study underpinned an undeniable difference of fecal microbiome between healthy and diseased dogs.

To: The results of the study identified several bacterial groups that differed significantly in the fecal microbiome between healthy and diseased dogs. 

From: If the observed variations predispose dogs to hip and elbow arthritis or if they are just a consequence, deserves further investigations.

To: If the observed differences in fecal bacterial composition predispose dogs to hip and elbow arthritis or if these differences reflect a correlation with these conditions deserves further investigation.

Introduction:

From: It has been also demonstrated, both in human and animal model, that the microorganism inhabitant the gut and the compounds they produce, as short chain fatty acids, lactic acid and metabolites of amino acids, affect host physiology.

To: It has been also demonstrated, both in human and animal models, that the microorganism inhabitants of the gut and the compounds they produce, such as short chain fatty acids, lactic acid, and metabolites of amino acids, affect host physiology.

From: Recent studies brought to light a possible new interplay of gut microbiome with pathological conditions, other than inflammatory bowel diseases, as metabolic syndrome and joint damage, the latter caused by chronical low-grade systemic inflammation.

To: Recent studies brought to light a possible new interplay of gut microbiome with pathological conditions, other than inflammatory bowel diseases, such as metabolic syndrome and joint damage, the latter caused by chronic low-grade systemic inflammation.

From: This pathology is multifactorial and is mainly related to genetic predisposition, mechanical events, age, sex, lifestyle and overweight.

To: This pathology is multifactorial and is mainly related to genetic predisposition, mechanical events, age, sex, lifestyle, poor body condition, and excess body weight.

Material and Methods

Animals and Housing

From: Throughout the period of study, dogs were housed in the usual home and condition and were followed by the owners and an informed consent was obtained from the owners of the dogs recruited for the study and.

To: Throughout the period of study, dogs were housed in the usual home and condition and were followed by the owners and an informed consent was obtained from the owners of the dogs recruited for the study and.

More information on the dogs in the study should be provided in this section. When I looked at the supplementary information, I found that the two groups differ significantly in body weight (t-test, P<0.01) with the AD dogs weighing more than the dogs in the HD group. Might this either play a role in the development of osteoarthritis or correlate with a reduction in activity?

Statistical analysis

Were the significance values adjusted for multiple comparisons in the analyses of the biochemical parameters? 

How were the reads normalized? Why are they described as normalized to 0/00 abundances rather than proportions or %? Did the sample groups differ in read counts? What was the minimum number of reads per sample? Were the samples rarified? 

Were differences in bacterial community composition tested at the family level as well as the genus level?

It is the Shannon Diversity Index or Shannon Index, not the Shannon Biodiversity Index, in the alpha diversity section. Similarly, when discussing the PCoA results, the authors should refer to beta diversity rather than beta biodiversity.

The lefse results are interesting. What are the relative abundance values for these taxa? Can you provide these in a table?

Discussion: 

page 8, line 259 Please correct Irritable Bowel Disease to Inflammatory Bowel Disease

page 9, line 322 Good point that the increase in Megamonas may reflect a decrease in other genera. Might there be a relationship with the decrease in Phascolarctobacterium?

page 10, line 342: Not clear what is meant here or if you showed this

Figure 4 on page 10: This figure on your hypothesis for infectious arthritis could be improved by describing how you would test it and which measurements would be needed. It could be used to support a future directions section but doesn't really relate to what you have done in this study.

Suggested changes to the supplementary text are provided in the attached document, including some confusion about the words arthritis and arthrosis.

Author Response

Responses to reviewer 1

We would like to thank the reviewer for their constructive comments.  The queries raised (Q) have been responded to Authors below.

In this study on the microbiome of dogs with and without osteoarthritis, the authors test whether differences were observed in a range of blood biochemistry results and the composition of gut (fecal) bacteria. This is a relevant topic in veterinary medicine and the authors have some interesting preliminary findings with their study of a relatively small number of dogs (n=13 or 14 per group).

Q: The main thing that should be addressed in a revision is that the authors hypothesize that a leaky gut barrier may play a role in the development of osteoarthritis. One issue with their hypothesis as stated is that the authors are not effectively describing how the loss of barrier integrity or increased epithelial permeability (“leaky gut”) is allowing increased interaction of MAMPs with the innate immune system. Their use of “microorganisms” - which a reader would assume means viable whole microbes and therefore systemic dissemination is confusing. Most of the current literature (predominantly human) link aberrant immune reactions (Th17) leading to immune mediated osteoarthritis, rather than infectious arthritis. If they intend to propose and test a potential mechanism for the development of osteoarthritis that involves bacteria crossing the barrier and translocating to the joints, they need to test for microbes/microbial molecules in the affected joint tissue. Unless they document microbial translocation to the joints or can otherwise support their claims, they need to revise their hypotheses. For example - if the arthritic dogs have serum IgG specific to the implicated fecal bacteria.

Authors: We agree with the comment of the reviewer and changed the text (lines 47-53) since not having investigated molecules that affects joint degeneration, it turns out to be too risky.

I have some questions about the statistical analyses. In addition, there are some minor spelling/grammatical corrections that should be made to the manuscript. Some are listed in the suggestions below.

Q: In the abstract, I recommend modifying this sentence:

Abstract:

From: The results of the study underpinned an undeniable difference of fecal microbiome between healthy and diseased dogs.

To: The results of the study identified several bacterial groups that differed significantly in the fecal microbiome between healthy and diseased dogs.

Authors: we changed the sentence as suggested (lines 21-23)

From: If the observed variations predispose dogs to hip and elbow arthritis or if they are just a consequence, deserves further investigations.

To: If the observed differences in fecal bacterial composition predispose dogs to hip and elbow arthritis or if these differences reflect a correlation with these conditions deserves further investigation.

Authors: we changed the sentence as suggested (lines 23-25).

Introduction:

From: It has been also demonstrated, both in human and animal model, that the microorganism inhabitant the gut and the compounds they produce, as short chain fatty acids, lactic acid and metabolites of amino acids, affect host physiology.

To: It has been also demonstrated, both in human and animal models, that the microorganism inhabitants of the gut and the compounds they produce, such as short chain fatty acids, lactic acid, and metabolites of amino acids, affect host physiology.

Authors: we changed the sentence as suggested (lines 31-33).

From: Recent studies brought to light a possible new interplay of gut microbiome with pathological conditions, other than inflammatory bowel diseases, as metabolic syndrome and joint damage, the latter caused by chronical low-grade systemic inflammation.

To: Recent studies brought to light a possible new interplay of gut microbiome with pathological conditions, other than inflammatory bowel diseases, such as metabolic syndrome and joint damage, the latter caused by chronic low-grade systemic inflammation.

Authors: we changed the sentence as suggested (lines 41-43).

From: This pathology is multifactorial and is mainly related to genetic predisposition, mechanical events, age, sex, lifestyle and overweight.

To: This pathology is multifactorial and is mainly related to genetic predisposition, mechanical events, age, sex, lifestyle, poor body condition, and excess body weight.

Authors: We modified the sentence as suggested (lines 56-58).

Material and Methods

Animals and Housing

From: Throughout the period of study, dogs were housed in the usual home and condition and were followed by the owners and an informed consent was obtained from the owners of the dogs recruited for the study and.

To: Throughout the period of study, dogs were housed in the usual home and condition and were followed by the owners and an informed consent was obtained from the owners of the dogs recruited for the study and.

Authors: We do not see any difference from the original sentence to the new one suggested from you, anyway we did modify the sentence eliminating the “and” that was at the end of the sentence (line 84).

Q: More information on the dogs in the study should be provided in this section. When I looked at the supplementary information, I found that the two groups differ significantly in body weight (t-test, P<0.01) with the AD dogs weighing more than the dogs in the HD group. Might this either play a role in the development of osteoarthritis or correlate with a reduction in activity?

Authors: Certainly, an excess of weight represent a predisposing factor in the onset of osteoarthritis and a reduction of activity can lead to overweight. Unlikely, BCS scores were not recorder from the veterinary but we have implemented the Supplementary Table 1 with means, standard deviation and significance. The unique evaluation of overweight dogs that we can get from these results is the comparison to the ideal live weight of the standard of the breed (although live weight is not really a standard for a breed). Probably the two groups had a significant live weight since the breed, the age and the sex were differently represented.

Statistical analysis

Q: Were the significance values adjusted for multiple comparisons in the analyses of the biochemical parameters?

Authors: The significance values were adjusted for multiple comparison (lines 144-145).

Q: How were the reads normalized? Why are they described as normalized to 0/00 abundances rather than proportions or %? Did the sample groups differ in read counts? What was the minimum number of reads per sample? Were the samples rarified?

Authors: The reads were normalized as a proportion within each taxon from the total reads count per sample and multiplied by 1000. It is a matter of simplicity and better understanding of the resulting number, since some taxa presents very low abundance. Total read count did not differ between groups and the minimum number of total reads were reported in the text (lines 133-134). Samples were not rarified but low represented taxa were eliminated from the analyses, as indicated in the text (lines 136-137).

Q: Were differences in bacterial community composition tested at the family level as well as the genus level?

Authors: We computed Shannon and Evenness indices at family level as suggested and Figure 1 was modified (page 6). Even beta diversity at family level was computed and reported in Figure 2 (page 6). Accordingly, we modified the text in the result and discussion sections (lines 174-178; lines 298-302).

Q: It is the Shannon Diversity Index or Shannon Index, not the Shannon Biodiversity Index, in the alpha diversity section. Similarly, when discussing the PCoA results, the authors should refer to beta diversity rather than beta biodiversity.

Authors: We changed biodiversity in diversity, as suggested.

Q: The lefse results are interesting. What are the relative abundance values for these taxa? Can you provide these in a table?

Authors: We added to the Supplementary Material the tables relative to family and genus (Table S3 and Table S4, respectively).

Discussion:

Q: page 8, line 259 Please correct Irritable Bowel Disease to Inflammatory Bowel Disease

Authors: Corrected (line 287).

Q: page 9, line 322 Good point that the increase in Megamonas may reflect a decrease in other genera. Might there be a relationship with the decrease in Phascolarctobacterium?

Authors: We provide an explanation directly in the discussion and we added references [68-69] we found to support this hypothesis (lines 327-332).

Q: page 10, line 342: Not clear what is meant here or if you showed this

Author: We have rewritten the sentence (lines 352-354). We hope that now is clearer.

Q: Figure 4 on page 10: This figure on your hypothesis for infectious arthritis could be improved by describing how you would test it and which measurements would be needed. It could be used to support a future directions section but doesn't really relate to what you have done in this study.

Authors: We deleted the figure since it did not relate well to the topic of the article and we modified the last part of the discussion (lines 358-361).

Q: Suggested changes to the supplementary text are provided in the attached document, including some confusion about the words arthritis and arthrosis.

Authors: We revised the supplementary text and words.

Reviewer 2 Report

Interesting work here. A scientific editor should be employed here to rearrange grammar and sentence structure of this paper. I would have liked to have seen Body Condition scores included for the test animals as perhaps your findings may have been due to obesity. 

Had you considered looking at not just CRP but also serum cartilage degeneration markers ( CTX-II for example) only because CRP is driven by any inflammatory process? There is clearly a difference between healthy and diseased dogs with respect to the microbiome but I am not sure that translates into a marker for OA but you did say that.

Author Response

Responses to reviewer 2

We would like to thank the reviewer for their constructive comments.  The queries raised (Q) have been responded to Authors below.

Q: Interesting work here. A scientific editor should be employed here to rearrange grammar and sentence structure of this paper. I would have liked to have seen Body Condition scores included for the test animals as perhaps your findings may have been due to obesity. 

Authors: Grammar and sentence structure were revised. We agree with your advice but unlikely, we do not have the BCS list because veterinary did not provide us. The unique evaluation of overweight dogs that we can get from these results is the comparison to the ideal live weight of the standard of the breed (although live weight is not really a standard for a breed).

Q: Had you considered looking at not just CRP but also serum cartilage degeneration markers (CTX-II for example) only because CRP is driven by any inflammatory process? There is clearly a difference between healthy and diseased dogs with respect to the microbiome but I am not sure that translates into a marker for OA but you did say that.

Authors: Thank you for the suggestion, CTX-II could be an interesting marker. However, we sent samples to an external lab clinic and we could not retest blood samples for this parameter.